# Factors Driving the Workplace Well-Being of Individuals from Co-Located, Hybrid, and Virtual Teams: The Role of Team Type as an Environmental Factor in the Job Demand–Resources Model

**DOI:** 10.3390/ijerph20043685

**Published:** 2023-02-19

**Authors:** Jaroslaw Grobelny

**Affiliations:** Faculty of Psychology and Cognitive Science, Adam Mickiewicz University, 60-568 Poznań, Poland; jaroslaw.grobelny@amu.edu.pl

**Keywords:** workplace well-being, remote work, hybrid work, virtual teams, hybrid teams, job demand–resources

## Abstract

(1) Background: An essential task for public health and industrial and organizational psychology specialists is maintaining employees’ workplace well-being. This has become more difficult with pandemic-induced changes (i.e., the shift to remote work and the rise of hybrid teams). This research adopts a team perspective to explore the issue of workplace well-being drivers. It is hypothesized that the team type (co-located, hybrid, or virtual) should be recognized as a unique environmental factor, resulting in the need for different resources for members of these teams to maintain their well-being. (2) Methods: A correlational study was conducted to systematically compare the relationship (its significance and importance) of a wide range of demands and resources with the comprehensively measured workplace well-being of members of co-located, hybrid, and virtual teams. (3) Results: The results confirmed the hypothesis. The significant drivers of well-being in each team type were different, and the ranking of the most important drivers within each team type varied. (4) Conclusions: Team type should be considered a unique environmental factor, even for individuals from different job families and organizations. This factor should be considered in practice and research employing the Job Demand–Resources model.

## 1. Introduction

Workers’ well-being has recently become one of the most significant areas of interest for both industrial and organizational (I/O) psychologists and public health specialists. The subject has gained such recognition that researchers are convinced that increasing employees’ well-being, an essential component of mental health, should be considered valuable for its own sake [1]. Well-being promotion was sanctioned as crucial for societies by the United Nations and thus included in its Sustainable Development Goals [2]. However, measures to improve well-being became more difficult after the outbreak of COVID due to the lasting changes in workplaces brought about by the pandemic. This paper aims to address these difficulties by increasing our understanding of the factors that drive the well-being of individuals working in different types of teams. In doing so, the paper contributes to the I/O psychology and public health literature on three counts: (1) it systematically compares the relative importance of a series of workplace well-being drivers across members of three different team types based on a large, multi-national sample; (2) it offers a team-level perspective in explaining employees’ well-being; and (3) it extends the Job Demand–Resource (JD–R) model by proposing the inclusion of team type as a unique environmental factor.

### 1.1. Workplace Well-Being and Remote and Hybrid Work

Since positive psychology has gained widespread recognition, there has been a shift in workplace-related health practices [3]. In place of the “deficit model”, researchers and practitioners began emphasizing the resources necessary for a workforce’s health. Consequently, well-being—defined as a state of optimal functioning and experience—has become the center of psychologists’ and public health specialists’ attention. One can distinguish a subjective (hedonic) perspective in which well-being is associated with happiness, positive emotions, and a lack of negative emotions, or a subjective (eudaimonic) perspective, wherein well-being is based on the purposeful and meaningful actions of individuals [4]. Both aspects are essential for employees’ health. However, hedonic well-being refers to personal judgment on satisfaction with particular life domains (such as work) and fulfilling one’s desires [5]. This state can be shaped by the employer’s actions involving allocating resources to individuals and teams. For this reason, the level of hedonic well-being can be shared to some extent by members of the same team. Meanwhile, the core parts of eudaimonia include the individual meaning of life and work, which are much more complex and grounded in basic psychological processes at a personal level. Thus, it seems that the first of the perspectives mentioned above is more suited to studying the resources and demands of team members. Hence, this paper adopted a hedonic view; thus, well-being at work is described as a state of positive job-related opinions, positive emotions, and a lack of negative emotions resulting from a person’s work. 

One of the most distinctive changes brought about by the 2020 COVID pandemic was the prompt shift in a substantial portion of the workforce to remote work (or “telework”). As many as 557 million people worked remotely during the pandemic, accounting for more than 17% of the global labor pool [6]. This number most likely increased in 2022 [7]. Until recently, remote work was commonly defined as working on a company’s premises for at least one day per week [8]. However, this definition was challenged when academics and practitioners noticed that people working remotely only one day per week versus an entire week differed significantly in their experiences [9]. The term “hybrid work” is presently used more often to describe a blended model in which employees combine an in-office presence and telework [10]. The distinction and differences between fully remote and hybrid work types have yet to be recognized, as the latter is a new and unexamined phenomenon [11]. 

The understanding of remote and hybrid work and employees’ well-being, even at the individual level, is limited. Some studies suggest positive outcomes of these flexible work arrangements, such as lower exhaustion (including emotional exhaustion) and stress, more positive and fewer negative work-related emotions, and higher job satisfaction [12,13,14,15,16,17]. At the same time, other studies suggest that remote and hybrid workers experience more negative emotions, emotional exhaustion, and cognitive stress than in-office employees [18,19]. Moreover, some suggest mixed or non-significant outcomes [20], or even that a curvilinear relationship exists between the number of remote work hours and employees’ well-being [15,21].

### 1.2. Team Perspective and Well-Being Drivers

The combination of team members’ working modes mentioned above will lead to significant group-level consequences and the emergence of different team types. A team can be co-located, meaning that every member works from an exact location, or virtual, meaning that each member works remotely. When people work both onsite and remotely within a single unit, this is referred to as a hybrid team [22], which seems to be a unique entity, not merely a mix of co-located and virtual members. Researchers have reported that the members and leaders of such teams face specific demands, such as different communication patterns, relationship breakdowns, and trust issues [23,24]. The presented classification was adapted for this paper, as it simultaneously captures several substantial differences between various teams. The listed team types differ in their dispersion and distance, be it physical, perceived, or psychological [25,26]. Achieving their shared goals requires communication through media of different richness and opposite models (i.e., synchronous and asynchronous). Members of these teams experience distinct challenges in building trust and sharing responsibility [27]. Finally, such classification is further justified by the need to provide reliable knowledge within this field. In recent years, many teams have become remote or hybrid overnight, which has created tremendous difficulties for their leaders and members [28].

Workplace well-being is rarely studied at the team level (i.e., by comparing the opinions and emotions of members of different types of teams) [29]. Even so, the hybrid team entity is particularly unexplored, as its emergence and spread are relatively recent [11,30]. Moreover, team type is often considered a driver (i.e., a factor leading to increases or decreases) of workplace well-being. As proven by the mixed results described above, this may not necessarily be true. Instead, some researchers have focused on determining what factors (e.g., job resources) determine the workplace well-being of members of a specific type of team. For example, Xiao et al. [31] examined a broad range of factors (from psychological to team- or family-related) for people working on remote teams. However, knowledge in this area is still severely limited, with studies systematically comparing different types of teams almost non-existent [11].

### 1.3. Hypothesis and Research Question Development

The author drew on the JD–R model to hypothesize that the team should be considered a unique environmental factor rather than a specific job resource or demand factor. A core assumption of the JD–R model is that working in a particular occupation or organization involves specific risks and challenges that result in different demands. As such, people in different professions or organizations require a specific set of resources to maintain their well-being [32,33]. Research within the JD–R model consequently focuses on occupations or types of companies, neglecting the team as an environment. Team type can drastically change group dynamics by altering communication patterns, work organization, interpersonal relationships, and trust. Notably, this factor is frequently beyond an individual’s control. Even if an employee persists with a traditional work routine, if even one team member shifts to remote work, they become a hybrid team. As such, the team type itself can constitute a distinctive environment, affecting its members regardless of their professions or work characteristics. Therefore, it was hypothesized that team type constitutes a unique environment shared by individuals from different professions and organizations. These environments impose distinct demands, requiring differing resources from members of each of the three team types. Hence, the difference between co-located, hybrid, and virtual team members is not in their well-being levels per se, but in their well-being drivers.

**H1.** *Team type (co-located, hybrid, or virtual) moderates the relationship between workplace well-being and its drivers*.

To test this hypothesis, a series of potential well-being drivers was selected. The intent was to reflect the broad spectrum of resource types described in the JD–R model and that has proven to be significant in prior research. Given that their relationship to the workplace well-being and health of stationary and remote individuals has previously been documented, the following personal resources were selected: trait emotional intelligence [34,35,36], healthy lifestyle [31,37,38], household members [31], and workstation [39,40,41]. Job resources included a flexible work schedule [42,43,44], team and team leader relations and communication quality [19,45,46,47,48,49,50], team and team leader communication frequency [21,31,51,52,53], and communication layers [11,21,54,55,56]. All of these factors were incorporated into the further analyses intended to test the stated hypothesis.

Assuming that the differences described in the hypothesis occur, it will be crucial to consider which of these drivers has the highest importance for each team type member’s well-being. Consequently, a research question was formulated to determine the relative importance of each investigated job resource as a driver of workplace well-being for each team type. The weight of this question is considerable because, due to the lack of systematic comparisons in prior research, it is unclear as to which factors may be important for employees’ well-being in general and which are only important for particular types of teams. This research question will be addressed by carrying out additional analyses using an exploratory approach.

## 2. Materials and Methods

A correlational study with workplace well-being, job demand, and job resources’ (well-being drivers) measurement was designed to gather the necessary data and compare co-located, hybrid, and virtual teams. A total of 1264 team members participated fully, the majority of whom came from the United States (*n* = 598) and Poland (*n* = 613). Others came from different countries (including Germany, New Zealand, India, The Netherlands, and others). The sample comprised 64.9% women, 32.6% men, and 2.5% who provided other responses or refused to answer. Participants were, on average, 35.1 years old (*SD* = 9.8), had 12.9 years of total job experience (*SD* = 10.1), and had 3.1 years of experience on their current teams (*SD* = 3.5). 

### 2.1. Sampling and Data Collection Procedure

The participants were recruited via direct and group-targeted messages on professional social media platforms. It is impossible to estimate how many of those who saw the invitation actually participated. However, the completion rate was 65.7%. The participants were included in the sample if they were employed, worked as a team member, had a nonmanagerial position, and performed work that could be (at least partially) done remotely. Inclusion criteria were checked by a series of screening questions, and 18.3% of those screened were excluded because they did not meet the inclusion criteria. No responses were excluded due to missing data.

Of the 1264 participants, 32.6% worked on co-located teams, 35.8% worked on hybrid teams, and 31.6% worked on virtual teams. These numbers were equally distributed between the two main countries. The even distribution of subjects from each team type was achieved by employing quota sampling. The intended sample size for each team was 400 people. Based on assumptions of medium effect size, the 22 predictors included, and an error probability of 0.01, the achieved statistical power was greater than 0.99 for each of the three samples, as calculated with Gpower (version 3.1.9.6.).

After an invitation, each participant was provided with a link to an anonymous survey. Informed consent and responses to every questionnaire and rating scale, along with demographic data, were collected. Team members were free to take part in the survey at the time and place of their choice. Data were collected from 24 May to 31 August 2022. Two attention check questions were used to ensure data quality.

### 2.2. Measurements

The series of tests and rating scales that were used to measure team members’ well-being, job demands, and resources included the following:Workplace well-being (cognitive component) [57,58]: The revised Job Descriptive Index (JDI) was used to evaluate the participants’ work-related opinions. Only the Job in General scale, consisting of 18 items, was administered. Participants were asked to rate whether a particular adjective (e.g., pleasant or bad) described their job. The results were reliable (*α* = 0.92).Workplace well-being (affective components) [59,60]: A 20-item version of the Job-related Affective Well-being Scale (JAWS) was used to measure both positive and negative emotional components of workplace well-being. On a 5-point scale, participants evaluated how often their jobs made them feel certain emotions during the last 30 days. The list comprises 10 positive (e.g., content) and 10 negative (e.g., angry) emotions, and two separate scores were calculated for each person. Both results had acceptable reliability (*α* = 0.83 for positive and *α* = 0.81 for negative).Demands and workload [61,62]: Four items assessing psychological demands and workload from the Demand–Control–Support Questionnaire (DSCQ) were adapted in this study (e.g., “Does your job require you to work very fast?”). Using a 7-point scale, participants had to evaluate how often their jobs demanded that they make specified efforts (e.g., working too fast). The scale had acceptable reliability (*α* = 0.82).Trait emotional intelligence [63]: A short version of the Trait Emotional Intelligence Questionnaire (TEIQue-SF) was used. Participants were asked how much they agreed with 30 statements about themselves using a 6-point scale (e.g., “I usually find it difficult to regulate my emotions”). Only a general score was calculated, and it was proven to be reliable (*α* = 0.85).Healthy lifestyle: The participants were asked to describe their healthy habits using three questions with 7-point Likert rating scales. The questions concerned their overall physical activity, physical exercise (frequency and intensity), and healthy-to-junk food ratio. The questions were based on those used by Xiao et al. [31].Household members: The participants were asked to report with whom they shared their households. A multi-choice question was used for this purpose, with the possibility of reporting at least one adult in the same household, a child (aged seven and older), a toddler or infant (aged six or younger), or no other people.Workstation: Five questions with regard to satisfaction from various aspects of participants’ workstations were asked. A 7-point Likert rating scale (from extremely dissatisfied to extremely satisfied) was used. The participants rated their workstation set-up’s quality, their surroundings’ visual components, air and thermal quality, noise density, and lack of distraction. The above aspects were selected based on a study by Xiao et al. [31].Flexible working hours: A single question with a 7-point Likert rating scale was used to determine how satisfied the participants were with the flexibility of their work schedule (from extremely dissatisfied to extremely satisfied).Team relations and communication quality [64]: Communication and cooperation within the work group subscale from the Work Group Characteristics Measure (WGCM) was used to evaluate this construct. The participants used a 7-point scale to rate how much the description of a well-performing group applied to their work team (e.g., whether members were willing to share information). The results were reliable (*α* = 0.85).Team leader relations and communication quality [65]: The leader–member exchange construct, describing mutual respect, trust, and obligation between a leader and their team member, was used to define this variable. Accordingly, the Leader–Member Exchange 7 questionnaire (LMX-7) was employed (the sample question was “How well does your leader understand your job problems and needs”), which proved to be reliable (*α* = 0.88).Team communication frequency and team leader communication frequency: Two questions with seven-point Likert rating scales (from never to very often) were used to evaluate how often participants communicated with peers from their teams and their team leaders.Communication layers: The participants were asked three questions describing the video, audio, and text layers (one question per layer), with a 7-point rating scale (from never to always). On all occasions in the last 30 days, they rated how often they communicated with their team colleagues while being able to see them (e.g., face-to-face or via online video communication services), only hear them (e.g., by phone or online with cameras off), or only read messages from them (e.g., via e-mails).

### 2.3. Statistical Analysis

The *R* language (version 4.1.2.) [66] with *RStudio* (version 2022.07.1 + 554, Posit, Boston, MA, USA) was used for all analyses. A common method variance analysis was performed as part of the preliminary results evaluation. Then, in the first step, a well-being score was estimated for each team member. A single factor was derived from three variables (cognitive and both negative scores) to operationalize workplace well-being in alignment with the theoretical basis. This was done with the R package psych [67]. Next, a series of multiple linear regression models was calculated by regressing the estimated well-being score on all workplace demand and resource measures. A general model and three sub-models, one for each team type, were analyzed. This aimed to determine the differences between co-located, hybrid, and virtual teams. Because job demands and resources tested within the scope of the JD–R model consistently proved to have additive, rather than joint or interactive, effects on health outcomes [33,68,69], all variables were entered into the regression with no interactions tested. Finally, a relative importance analysis was performed per Johnson’s relative weights analysis (RWA) [70]. This supplementary analysis allows one to determine the unique contribution that each independent variable from a regression model makes while predicting outcome variability. Regression coefficients cannot be relied upon in such matters when predictors are correlated, which is a common case for workplace resources [71]. Both analyses aimed to identify uniquely significant and important well-being drivers for each team type. 

## 3. Results

The study results are summarized in Table 1, and Appendix A provides information on the correlation between all variables and their means across team types. Since all of the data was collected using self-reports, the results may be susceptible to a common method variance (CMV) issue. A strategy presented by Williams and McGonagle [72] was employed to investigate it. Unmeasured latent method construct (ULMC) was analyzed in the first phase. A baseline model that includes all substantive variables (items with their corresponding constructs as latent variables) was fitted using confirmatory factor analysis. A second model was then fitted, adding method factor loadings (MFLs) to each item. MFLs were allowed to be freely estimated. Method variable loadings and error variance in the baseline model were fixed to corresponding values from the second model. Both models were tested using the χ^2^ difference test, proving to be significantly different (df_1_ = 1401; df_2_ = 1449; χ^2^_1_ = 10,426; χ^2^_2_ = 10,784; *p* < 0.05). However, this should be expected due to the large sample size. More importantly, only 2.2% of the total variance was attributed to the method construct. Based on this, no significant CMV was concluded, and no further testing was conducted.

Factor analysis was performed to estimate each team member’s comprehensive well-being score. Three variables were included: JDI, JAWS-positive, and JAWS-negative scores. All variables correlated significantly and strongly (absolute coefficient values were between 0.63 and 0.73). Bartlett’s test outcomes (*chi*^2^ = 1752.49; *df* = 3; *p* < 0.001) and a KMO measure of 0.72 suggested that the selected variables were adequate for dimension reduction. The maximal likelihood factoring method was selected. The model presented an exceptional fit to the observed data (*chi*^2^ = 1756.43; *p* < 0.001; *AGFI* = 0.99; *TLI* = 0.99; *CFI* = 0.99; *RMSEA* = 0.01), and the single factor explained 78% variability of the input variables. Loadings for all variables were high and comparable (0.85 for JDI, 0.86 for JAWS-positive, and −0.74 for JAWS-negative scores). Factor scores were estimated using the regression-based weight approach [73] to value workplace well-being ratings; they were then passed for further analysis. 

Next, the estimated well-being scores were regressed on a set of predictors (i.e., demands and workload, trait emotional intelligence, healthy lifestyle, household members, workstation satisfaction, flexible working hours, team and team leader relations and communication quality, team and team leader communication frequency, and communication layers). An overall model was prepared, followed by three secondary models, one for each team type. All models proved significant and explained a substantial part of the team members’ well-being variability, as presented in Table 2. An advanced diagnostic and model residuals analysis indicated no issues, such as heteroskedasticity (Goldfeld–Quandt test) or the autocorrelation of models’ errors (Durbin–Watson test), and a linear fit of the models was adequate (Rainbow test). Since the regression models were prepared on data from convenience samples, there was a bias risk due to over- or under-representation of specific team member sub-groups. To assess that risk, a k-fold cross-validation was performed [74]. Each team-type sample was shuffled and split into five subsamples (k = 5). Then, in five iterations, all subsamples but one were used to build a regression model whose fit was tested on the remaining part (different in each iteration). Error measures and coefficients of determination were computed for each repetition. Should the results be biased due to the sample’s composition, these outcomes would be significantly lower than the original models’ parameters discussed above, and considerably dispersed. As presented in Table 2, this was not the case.

Regression coefficients varied greatly between team-level models in terms of their significance, as presented in Table 3. This supports H1. Team type constitutes a unique environmental factor, resulting in the need for different resources for team members’ well-being. Only a few variables predicted well-being significantly for co-located, hybrid, and virtual team members altogether (i.e., flexible working hours, leader–member exchange, frequency of video-based communication, having a young child as a household member, and workstation air and thermal quality). For each team, a specific set of significant well-being drivers can be identified (e.g., emotional intelligence, team communication frequency, and audio-based communication for co-located team members). Notably, in some cases, one team’s job resource was another team’s demand (e.g., team communication frequency). 

Finally, RWA was conducted to estimate how much each variable contributed to team members’ well-being, given that they were all correlated. Table 3 presents ε coefficients, which indicate how much of the explained well-being variance can be attributed to each predictor. The wide dispersion of these values between teams indicates that members of different team types require different resources due to the unique characteristics of their environment, further supporting H1. Notably, the ranking of variables making the highest contributions (i.e., the most important well-being drivers) for members of each team type varied noticeably (except for the very first variable in the rankings, i.e., flexible working hours). For co-located teams, the most important drivers were flexible working hours, leader–member exchange, and a series of workstation-related factors, such as workstation set-up quality. However, for hybrid teams, the most important drivers were team leader communication frequency and team relation quality, but none of the workstation-related variables appeared in this ranking. Furthermore, for virtual teams, factors related to household members and communication layers proved to be more important to some extent. The results of the RWA allowed for the preparation of lists of significant and unique resources, providing the answer to the explorative research question.

## 4. Discussion

This study aimed to investigate whether team type is a unique environmental factor moderating the significance and importance of workplace well-being drivers. In general, the collected data supported the hypothesis. Depending on the team type, specific job resources proved to have different relationships with team members’ well-being (in terms of significance and importance). While factors such as flexible working hours and leader–member exchange were found to be universally important for well-being, lists of subsequent important drivers varied considerably within each team type. For co-located team members, this list somewhat surprisingly included workstation-related factors. Offices tend to be equipped to a high standard (often determined by state labor law), and equipment or workplace quality issues are typically attributed to remote work [41]. Virtual team members (who rated their workstation satisfaction in the current study as higher) typically have more autonomy in designing their workstations than their onsite colleagues. Previous research has confirmed that workstation ergonomic interventions performed without employee involvement demonstrate little or no success in improving worker health [75]. The current findings suggest that autonomy and decision-making, rather than mere equipment in the workstation domain, could be considered a key driver of workplace well-being. Interestingly, despite its previously proven significance [36], a psychological resource, such as emotional intelligence, was found to be important for well-being only in co-located teams. Arguably, effective conflict resolution in co-located teams is a more common demand [76,77]. Hence, the relative role of the corresponding resources in the other two types of teams was minor.

The well-being drivers’ analysis of virtual team members revealed the utmost importance of the team leader role and the communication layers. The meaning of leadership in a remote context has already been postulated within the JD–R model [78]. As such, the findings confirm the assumption of team leaders’ considerable role. The leaders’ communication is commonly limited to text and audio due to the characteristics of the virtual work environment. The findings on remote work and social isolation are still inconclusive [17], but teleworkers have reportedly perceived a lack of face-to-face communication [52]. Incorporating video-layer communication in working with these teams could be essential in preventing feelings of isolation and maintaining well-being. Notably, the frequency of communication alone may not be sufficient to increase virtual team members’ workplace well-being, contrary to previous findings [31,79]. As the results show, when the classification of communication with the entire team versus only with the leader is considered, only the latter is a relevant well-being driver.

The relatively high importance of hybrid team communication quality supports previous research suggestions about trust issues between these teams’ onsite and remote sides [21,23,47]. However, as this factor proved to be a significant driver of workplace well-being for virtual team members, trust issues are likely to affect those working in the same remote manner. Although the results showed some similarities in the well-being drivers of hybrid team members to those of virtual teams, there were also noticeable differences. For example, a lack of distractions or household members were much more important in hybrid teams. A unique feature of this team type as a work environment included the incomparably high importance of work flexibility. The weight of this factor for teams and participants in hybrid meetings has previously been highlighted in the literature [11]. The current research’s findings support the notion that flexibility is a leading driver of workplace well-being among hybrid team members. 

### 4.1. Theoretical and Practical Implications

The current study’s results contribute to public health and I/O psychology theory and practice, and can be put into the broader context of the JD–R model. Until now, job families and organizations have primarily been recognized and studied as context factors [33]. Recently, the JD–R model’s authors recognized the urgency of including group-level analysis when discussing workers’ well-being in a pandemic crisis [78]. However, they only examined specific processes within a worker’s team or family rather than completing a systematic comparison of different settings. An updated perspective can be proposed based on the analyzed results. Arguably, the type of team on which one works should be considered an environmental factor rather than a resource itself. The consequence of this approach should be to understand co-located and virtual teams as different environments, one that, even within the same profession or organization, will pose different demands on its members. Consequently, in line with the confirmed hypothesis, team members require differentiated resources to maintain their workplace well-being. Thus, researchers and practitioners who employ the JD–R model should also expand it to include the type of team on which the people affected by their research and interventions work. Notably, such an extension aligns with future research suggestions to include group-level factors that were recently formulated for the JD–R model [80]. Moreover, the presented study provides evidence supporting the significance of resources that have not been previously tested within the JD-R model and that are beyond the postulated scope of its development. For example, tangible resources such as ICT tools or proper rewards were mentioned as a means to cope with the psychological strain on team members during the recent health crisis [78]. However, the current study suggests that for some teams (i.e., hybrid and virtual), communication and relationship quality factors have a relatively substantial effect.

The results also contradicted several widely held beliefs and myths about remote and hybrid work, the scientific status of which are unclear. Similar to the performance case, a surprisingly positive impact of the presence of other household members was identified for members of hybrid and virtual teams [81]. However, a healthy lifestyle, frequently the subject of organizational interventions led by health professionals and HR departments, proved unexpectedly irrelevant in every team type. The limited effectiveness of such interventions was previously suggested [37]. The findings confirm the relatively low importance of a healthy lifestyle compared to other factors, such as leadership and communication. Notably, the subject under consideration is workplace well-being, not well-being in general. Such detailed study outcomes, which address the research question posed, should form the basis of evidence-based practice by public health experts and I/O psychologists.

### 4.2. Limitations and Future Guidelines

As with all studies, the present study has several limitations, despite the best efforts to ensure high research quality. The obtained sample could raise concerns. As is often the case in online research, an equal distribution of gender and age groups was not achieved. The participants mainly came from two countries representing Western culture. Regarding the studied variables, much more emphasis was placed on the resource side and the positive aspects of health and well-being, which were derived from the adapted hedonistic approach. It can be speculated that this caused a surprising lack of significance of job demands in the computed models. Finally, the study employed a concurrent measurement of predictors and outcomes, which limited the possibility of drawing long-term conclusions. A repeated study with a more diverse participant group (and purposive sampling plan), one that includes multi-source measurement of a broader range of job demands and negative aspects of health (e.g., burnout), and a delayed criterion measurement would increase the reliability of the results and further confirm the conclusions. Finally, the implied direction of the relationship, i.e., resources affecting workplace well-being, is derived from the theoretical framework of the JD-R model. Further experimental research would empirically confirm the status of this link.

## 5. Conclusions

Addressing workers’ well-being is an ongoing challenge in the post-pandemic era. A vital factor for consideration in research and practice in this area should be the type of team in which an individual works, which can be considered as one of the essential environmental factors. As such, members of co-located, hybrid, and remote teams differ in their need for job resources that drive their workplace well-being. These three types of teams, which have not previously been systematically compared and studied in the context of the JD-R model, can form the basis for incorporating a team-level approach into JD-R theory.

## Figures and Tables

**Table 1 ijerph-20-03685-t001:** Descriptive statistics.

Variable	*M* 95% CI [LL, UL]	*SD*	Range (*Spread*)	*W*	*Skew*	*Kurt*
JDI	2.52 [2.49, 2.54]	0.44	2 (66%)	0.89 ***	−1.13	3.81
JAWS-positive	3.05 [3.01, 3.10]	0.84	4 (80%)	0.98 ***	−0.13	2.52
JAWS-negative	2.70 [2.66, 2.74]	0.77	4 (80%)	0.98 ***	0.40	2.82
DCSQ	4.21 [4.16, 4.25]	0.90	5 (71%)	0.98 ***	0.39	3.05
TEIQue-SF	3.76 [3.73, 3.79]	0.51	3.33 (66%)	0.97 ***	−0.55	3.70
Healthy lifestyle: Physical activity	3.78 [3.70, 3.86]	1.46	6 (85%)	0.94 ***	−0.02	2.80
Healthy lifestyle: Physical exercise	3.39 [3.30, 3.48]	1.63	6 (85%)	0.93 ***	0.11	2.14
Healthy lifestyle: Healthy food intake ratio	4.75 [4.68, 4.81]	1.12	6 (85%)	0.92 ***	−0.40	3.20
Household members: 1+ adult	81% [78%, 83%] ^a^	-	-	-	-	-
Household members: 1+ child (7–18 y)	20% [18%, 23%] ^a^	-	-	-	-	-
Household members: 1+ child (0–6 y)	15% [13%, 17%] ^a^	-	-	-	-	-
Household members: no other person	17% [15%, 19%] ^a^	-	-	-	-	-
Workstation: set-up quality	4.88 [4.79, 4.97]	1.60	6 (85%)	0.92 ***	−0.64	2.75
Workstation: visual component	4.86 [4.77, 4.95]	1.63	6 (85%)	0.92 ***	−0.55	2.64
Workstation: air and thermal quality	4.92 [4.83, 5.00]	1.59	6 (85%)	0.93 ***	−0.55	2.64
Workstation: noise density	4.83 [4.73, 4.92]	1.74	6 (85%)	0.92 ***	−0.50	2.29
Workstation: lack of distraction	4.62 [4.53, 4.72]	1.65	6 (85%)	0.93 ***	−0.35	2.37
Flexible working hours	5.42 [5.32, 5.53]	1.82	6 (85%)	0.81 ***	−1.01	2.93
WGCM	5.69 [5.61, 5.76]	1.34	6 (85%)	0.86 ***	−1.12	3.97
LMX-7	5.21 [5.14, 5.29]	1.38	6 (85%)	0.93 ***	−0.85	3.36
Team communication frequency	5.43 [5.35, 5.51]	1.47	6 (85%)	0.88 ***	−0.81	3.06
Team leader communication frequency	5.24 [5.14, 5.33]	1.71	6 (85%)	0.87 ***	−0.86	2.81
Communication layer: video	3.45 [3.35, 3.55]	1.85	6 (100%)	0.91 ***	−0.28	1.84
Communication layer: audio	2.60 [2.50, 2.71]	1.87	6 (100%)	0.92 ***	0.31	1.89
Communication layer: text	2.62 [2.53, 2.72]	1.70	6 (100%)	0.93 ***	0.26	2.02

Notes. LL, UL—95% confidence intervals for means, lower and upper limits respectively; *Spread*—range spread (range divided by maximal value); *W*—Shapiro-Wilk test for normality; *Skew*—skewness; *Kurt*—kurtosis. ^a^ A formula for proportion confidence interval without continuity correction was used. *** *p* < 0.001.

**Table 2 ijerph-20-03685-t002:** Regression models’ analysis.

Statistics/Tests	All Participants	Team Type		
		Co-Located	Hybrid	Virtual
F	45.21 ***	22.19 ***	18.06 ***	14.12 ***
R^2^	0.44	0.56	0.48	0.45
Adj. R^2^	0.42	0.53	0.45	0.42
Goldfeld-Quandt test	0.86	0.62	1.16	0.75
Durbin-Watson test	1.88 *	1.92	2.00	2.05
Rainbow test	0.93	0.94	0.71	1.15
Cross-validation R^2^	0.42 (0.08) ^a^	0.52 (0.09)	0.42 (0.10)	0.41 (0.09)
Cross-validation MAE	0.60 (0.04) ^a^	0.57 (0.07)	0.59 (0.05)	0.53 (0.05)

^a^ Means (and standard deviations) from cross-validation’s five iterations. * *p* < 0.05; *** *p* < 0.001.

**Table 3 ijerph-20-03685-t003:** Regression and relative weight coefficients.

Variable	Team Type								
	Co-Located			Hybrid			Virtual		
	*B*	95% CI [LL, UL]	*ε*	*B*	95% CI [LL, UL]	*ε*	*B*	95% CI [LL, UL]	*ε*
(Intercept)	0.19	[−0.37, 0.74]	-	0.14	[−0.28, 0.56]	-	0	[−0.38, 0.37]	-
DCSQ	−0.01	[−0.09, 0.07]	12.29%	−0.06	[−0.14, 0.01]	29.87%	0.05	[−0.03, 0.12]	14.88%
TEIQue-SF	0.20 ***	[0.11, 0.29]	8.41%	0.04	[−0.03, 0.11]	16.35%	0.04	[−0.03, 0.11]	14.34%
HL: Ph. activity	0.09	[−0.02, 0.20]	6.51%	0.07	[−0.05, 0.19]	8.52%	0.07	[−0.03, 0.16]	11.49%
HL: Ph. exercise	0.01	[−0.10, 0.12]	2.10%	0.04	[−0.09, 0.16]	7.84%	−0.08	[−0.19, 0.03]	11.10%
HL: H. food ratio	−0.09 *	[−0.17, −0.01]	1.15%	0.01	[−0.07, 0.09]	5.27%	−0.01	[−0.09, 0.07]	10.31%
HH: 1+ adult	−0.36	[−0.91, 0.20]	9.45%	−0.31	[−0.72, 0.11]	4.34%	0.12	[−0.25, 0.49]	0.64%
HH: 1+ 7–18 y	0.16	[−0.03, 0.36]	7.12%	0.29 **	[0.10, 0.48]	4.18%	−0.11	[−0.29, 0.08]	3.24%
HH: 1+ 0–6 y	0.40 ***	[0.18, 0.63]	8.14%	0.32 **	[0.10, 0.53]	3.76%	0.49 ***	[0.28, 0.69]	3.75%
HH: no other	−0.27	[−0.86, 0.32]	1.71%	−0.26	[−0.72, 0.20]	3.32%	0.68 **	[0.28, 1.09]	1.91%
WS: set-up qual.	0.08	[−0.01, 0.18]	10.75%	0.04	[−0.05, 0.13]	2.96%	0.09	[0, 0.17]	1.00%
WS: vis. comp.	0.05	[−0.04, 0.14]	0.61%	−0.06	[−0.16, 0.04]	2.01%	−0.04	[−0.14, 0.07]	0.32%
WS: air and therm.	0.13 **	[0.04, 0.22]	5.34%	0.11 *	[0.02, 0.20]	1.93%	0.16 **	[0.04, 0.28]	7.79%
WS: noise density	0.05	[−0.04, 0.14]	1.53%	−0.07	[−0.18, 0.03]	1.86%	0.03	[−0.07, 0.13]	1.04%
WS: lack of dist.	0.17 ***	[0.07, 0.28]	7.06%	0.15 **	[0.06, 0.25]	1.80%	−0.11 *	[−0.20, −0.03]	0.65%
FWH	0.17 ***	[0.10, 0.23]	5.35%	0.34 ***	[0.26, 0.43]	1.60%	0.20 ***	[0.10, 0.30]	0.93%
WGCM	−0.22 ***	[−0.34, −0.10]	6.14%	0.12 *	[0.01, 0.23]	1.24%	0.17 **	[0.05, 0.29]	1.12%
LMX−7	0.19 **	[0.07, 0.31]	0.39%	0.25 ***	[0.13, 0.36]	1.13%	0.17 **	[0.05, 0.28]	0.50%
TC freq.	0.31 ***	[0.19, 0.43]	1.21%	−0.05	[−0.16, 0.06]	0.70%	−0.16 **	[−0.27, −0.05]	0.69%
TLC freq.	0.02	[−0.11, 0.15]	0.36%	0.05	[−0.06, 0.17]	0.50%	0.17 *	[0.04, 0.30]	5.21%
CL: video	0.11 *	[0.01, 0.21]	0.33%	0.17 ***	[0.09, 0.25]	0.38%	0.17 ***	[0.09, 0.25]	2.47%
CL: audio	0.14 ***	[0.07, 0.21]	0.65%	−0.02	[−0.10, 0.05]	0.27%	0.01	[−0.07, 0.09]	5.89%
CL: text	−0.02	[−0.11, 0.07]	3.38%	0.04	[−0.03, 0.12]	0.16%	−0.09 *	[−0.17, −0.02]	0.72%

Notes. HL—Healthy lifestyle; HH—Household members; WS—Workstation; FWH—Flexible working hours; TC—Team communication; TLC—Team leader communication; CL—Communication layer. *B*—standardized regression coefficients; 95% CI [LL, UL]—95% confidence intervals for *B* coefficients, lower and upper limits respectively; *ω*—relative weight coefficients. *p* < 0.1; * *p* < 0.05; ** *p* < 0.01; *** *p* < 0.001.

## Data Availability

The data presented in this study are openly available at https://osf.io/fudqj/ (accessed on 27 December 2022).

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
