# Peer review of "Factors Driving the Workplace Well-Being of Individuals from Co-Located, Hybrid, and Virtual Teams: The Role of Team Type as an Environmental Factor in the Job Demand–Resources Model"

_ijerph, 2023, doi:10.3390/ijerph20043685_

Round 1

Reviewer 1 Report

Comments to the Author

Review of the Manuscript:

 Thank you very much for providing me with the opportunity to review this manuscript Factors Driving the Workplace Well-being of Individuals from Co-Located, Hybrid, and Virtual Teams: The Role of Team Type as an Environmental Factor in the Job Demand–Resources Model. I found key features of the manuscript to be strengths, and these are 

a) The article touches upon and expands the literature on the team type and Job Demand-Resource.

b) The research goal is very relevant and important to employee well-being. 

Nevertheless, the study is subject to major limitations which I delineate below. I aim to provide suggestions with the hope that the authors will benefit from them in furthering this interesting manuscript and revealing its key strengths.

The following is a summary of the needed revisions.

Throughout the front part (sections 1.1 and 1.2), the authors discussed hedonic and eudaimonic perspectives and stated the paper is based on a hedonic view. The author should add discussion to justify why the hedonic view is selected.

The title of section 1.3 states “hypothesis and research question development,” however, no research question was discussed in this section.

In section 1.3, the author should add a discussion to justify why these three types of teams (co-located, hybrid, and virtual teams) were selected in this study. For example, what are the essences behind co-located, hybrid, and virtual teams? Could that be physical/psychological distance? Media richness? Prompt responses from other team members? Etc.

H1 is too broad and not clear as it only states “relationship between well-being and it drivers” – specific types of well-being and drivers should be included in the hypothesis. Also, it would be better to change the order by stating “relationship between drivers and well-being.”

In section 2.2, the author should add a brief description of each variable included, as well as sample item(s) for each variable.

The author might consider moving the section 2.3 to section 2.1 as both sections are about the sampling process.

In section 3, the author mentioned table S1 as well as multiple results from it. Since this table has been discussed extensively in the subsequent sections, the author should consider adding this table to the manuscript.

In the paragraph before table 1, the author stated that factor scores were calculated using regression. Please report detailed information about how these factor scores were calculated (e.g., independent and dependent variables that were included).

In the second paragraph of section 3, the author stated that well-being scores were regressed on a set of predictors. Please report the detailed information about the predictors. (what are they?)

Common method variance

The authors should consider using post-hoc method to test if CMV is an issue in the study. For example, the unmeasured latent method constructs (ULMC). Below is an example of CMV-related analysis.

Williams, L. J., & McGonagle, A. K. (2016). Four research designs and a comprehensive analysis strategy for investigating common method variance with self-report measures using latent variables. Journal of Business and Psychology, 31(3), 339-359.

Studies conducted.

This manuscript only includes one study, with online panel data and a single-source report. This is inadequate to support these hypotheses (not to mention part of the hypotheses were not supported). The authors should consider adding another study to replicate the results in the current study. 

I hope that the above comments help you improve the quality of your manuscript.

Good luck! 

Reviewer

Author Response

Thank you for your time and effort in reviewing my manuscript. I've found the comments both helpful and thoughtful. Please find my detailed responses attached. 

Reviewer 2 Report

Hello,

Please explain for your sample: the level of precision, the level of confidence or risk, and the degree of variability in the attributes being measured.

Obtaining a representative sample would contribute to a better explanation of the causal relationships and would allow to bring the evidence into the study.

Please find below my specific comments:

The study approaches team perspective to explore the issue of workplace well-being drivers. The team type (co-located, hybrid, or virtual) should be recognized as a unique environmental factor, resulting in the need for different resources for members of these teams to main tain their well-being.

Team type (co-located, hybrid, or virtual) moderates the relationship between well-being and its drivers.

The topic has been studied by other authors. The references to other studies are sufficient and clarify very well the developments in the field.

In a systematic approach the author tries to show that the only factor explaining well-being is related to team type (co-located, hybrid, or virtual)  

The sentence type of team in which an individual works, which is a unique environmental factor is too strong in relation to the working methodology and conclusions obtained.

The authors do not mention in the conclusions part what the study brings new. They limit themselves only to conclusions related to the subject analyzed without placing the results in a wider context.

The methodology is standard and well developed.

The objectives of the study are explicit. The conclusions are correctly formulated and correspond to the content of the scientific analyses carried out but are not placed in a wider scientific context.

Few problematisations are presented that can support future studies.

The bibliography is sufficiently developed.

Ethically the author mentions the limits of the research.

The online study has problems with the representativeness of the sample: distribution by country, type of organisation, gender.

Convenience sampling is only used when random and non-probability sampling - Purposive sampling is not possible

Causal inferences cannot be made using data from these types of samples.

This leads to false results.

Obtaining a representative sample would contribute to a better explanation of the causal relationships and would allow to bring the evidence into the study.

Author Response

(The authors gave the same response as above.)

Round 2

Reviewer 2 Report

Hello,

The author understood well what my reservations are about the sample construction.

As a sociologist working on representative samples I am always careful about their construction and how they can be tested whether they are probabilistic or not.

The arguments included in the revised version are convincing.

For new research, please take in consideration:

1.      Cochran, W. G. 1963. Sampling Techniques, 2nd Ed., New York: John Wiley and Sons, Inc.

2.      Israel, Glenn D. 1992. Sampling The Evidence Of Extension Program Impact. Program Evaluation and Organizational Development, IFAS, University of Florida. PEOD-5. October.

3.      Kish, Leslie. 1965. Survey Sampling. New York: John Wiley and Sons, Inc.

4.      Miaoulis, George, and R. D. Michener. 1976. An Introduction to Sampling. Dubuque, Iowa: Kendall/Hunt Publishing Company.

 Good luck in the future!